# Five Amino Acid Substitutions in the S1 Unit of Infectious Bronchitis Virus Are Critical Determinants Enhancing Its Adaptation to Vero Cells

**DOI:** 10.3390/vetsci12050394

**Published:** 2025-04-22

**Authors:** Zhichao Cai, Mingjing Zhang, Shouguo Fang

**Affiliations:** College of Agriculture, Yangtze University, Jingzhou 434025, China; caizhichao110@126.com (Z.C.); echozh@hotmail.com (M.Z.)

**Keywords:** infectious bronchitis virus, cell tropism, adaptation

## Abstract

The S1 subunit of the spike protein in avian infectious bronchitis virus (IBV) plays a crucial role in determining its host range, cellular tropism, and tissue specificity. Following the continuous passage of IBV-EP3 through Vero cells, the adaptability of IBV to Vero cells has progressively increased, leading to the accumulation of 19 amino acid substitutions in the S1 region of IBV-P65. In this study, we identify five specific amino acid changes as critical determinants that enhance the adaptation of IBV to Vero cells. Our findings provide valuable insights into the mechanisms underlying the adaptation of IBV to Vero cells and may facilitate the development of Vero cell-derived IBV vaccines by improving replication efficiency via targeted genetic modification.

## 1. Introduction

Coronaviruses are enveloped, single-stranded RNA viruses that infect a broad spectrum of mammalian and avian species [1]. According to phylogenetic and genomic analyses, coronaviruses are classified into four genera: *Alphacoronavirus*, *Betacoronavirus*, *Gammacoronavirus*, and *Deltacoronavirus* [2]. The viral genome encodes four primary structural proteins: the spike (S) protein, envelope (E) protein, membrane (M) protein, and nucleocapsid (N) protein. Among them, the S protein plays a crucial role in determining the viral host range, facilitating viral entry, dictating cell and tissue tropism, enabling cross-species transmission, and adapting to new hosts [3].

Infectious bronchitis virus (IBV), a member of the *Gammacoronavirus*, is prevalent globally in both industrial and backyard poultry populations [4,5,6]. Similar to other coronaviruses, the IBV S protein comprises two subunits: S1 and S2 [7]. The S1 subunit is responsible for receptor binding, whereas the S2 subunit mediates the fusion of the viral envelope with the host cell membrane [8,9]. The S1 subunit contains two domains: the N-terminal domain (NTD, amino acids 21–237) and the C-terminal domain (CTD, amino acids 269–414) [10]. One or both of these domains function as the receptor-binding domain (RBD) [11,12,13].

A multitude of diverse IBV strains circulate globally. The S1 subunit exhibits significant amino acid sequence diversity, whereas the S2 subunit remains relatively conserved across different strains [14]. Phylogenetic analysis of the S1 subunit has classified IBVs into six genotypes, encompassing 32 distinct viral lineages and a number of inter-lineage recombinants [15]. Naturally, IBV primarily infects the respiratory tract [16,17], while some strains can also infect additional tissues such as the kidney, oviduct, and gastrointestinal tract [4,18,19]. Different strains exhibit variations in tissue tropism, morbidity, mortality, pathogenicity, and production losses in poultry, which are believed to be attributed to amino acid variations in the S1 subunit [13,20,21].

Most field strains of IBV are incapable of infecting mammalian cell lines; however, the Beaudette strain represents an exception. The Beaudette strain was developed through serial passaging of the virulent Massachusetts M41 strain in embryonated chicken eggs and cultured cells. As a result of this adaptation process, the Beaudette strain has acquired the ability to infect baby hamster kidney cells (BHK-21), monkey kidney cells (Vero) [11,22], and even the human cell lines H1299 and Huh7 [23]. Research has demonstrated that the S-protein of IBV plays a crucial role in determining the extended host cell tropism of the Beaudette strain [7].

IBV-EP3 was generated by passaging the Beaudette strain through chicken embryonated eggs for three passages, while IBV-P65 was obtained by passing IBV-EP3 through Vero cells for 65 generations. During the serial passage through Vero cells, IBV gradually adapted to the Vero cells [24]. Compared with IBV-EP3, a total of 49 amino acid substitutions were identified in IBV-P65. Specifically, 26 of these substitutions occurred in the S protein, with 19 amino acid changes clustered in the S1 region. These mutations in the S protein likely expanded the host range of IBV [24].

In this study, using the established IBV reverse genetics system, six recombinant viruses that encompass the partial or total amino acid changes of the aforementioned 19 amino acid substitutions in the S1 subunit were generated by replacing the corresponding regions of IBV-P65 with those from IBV-EP3. Our findings indicate that the mutant virus, which replaces the 20,884–21,335 nucleotide sequence of the IBV-P65 S gene with the corresponding sequence from the IBV-EP3 S1 gene, exhibits significantly reduced and delayed growth kinetics as well as a lower replication efficiency in Vero cells compared to IBV-P65. Considering that only five amino acid substitutions (I181T, T246I, C267F, I273T, K296Q) differentiate this region between IBV-P65 and IBV-EP3, it is plausible to infer that these specific mutations in the S1 domain may enhance the adaptation of IBV to Vero cells.

## 2. Materials and Methods

### 2.1. Cell Culture and Virus

Vero cells (mycoplasma test negatively; passaged three generations after recovery) were cultured in Dulbecco’s Modified Eagle’s Medium (DMEM) (Invitrogen, Waltham, MA, USA) supplemented with 10% fetal bovine serum (FBS) (Hyclone, Logan, UT, USA), penicillin (100 units/mL), and streptomycin (100 μg/mL) (Invitrogen, USA) at 37 °C. IBV-EP3 (GenBank accession number DQ001338) is a Beaudette strain (GenBank accession number M95169.1) propagated in chicken embryonated eggs for three passages. IBV-P65 is a Vero cell-adapted IBV Beaudette strain (GenBank accession number DQ001339.1) obtained by passaging IBV-EP3 for 65 generations in Vero cells.

### 2.2. Generation of Mutant Viruses

Several fragments containing the sequences of IBV-P65 or IBV-EP3 were amplified by PCR and cloned into pGEM-T easy (Promega, Madison, WI, USA). The primers (synthesized by BGI, Shenzhen, China) and the obtained plasmids are detailed in Table 1 and Table 2, respectively. After confirmation via sequencing analysis, the plasmids were subjected to digestion with BsmB I or Bsa I, and the corresponding enzyme-digested fragments were subsequently purified. The full-length cDNA was assembled by substituting the corresponding fragment with the mutant fragment, as previously described [25]. The construction process of the recombinant viruses is illustrated in Figure 1. Full-length transcripts were synthesized in vitro by utilizing the mMessage mMachine T7 kit (Ambion, Austin, TX, USA) according to the manufacturer’s protocol. These transcripts were subsequently electroporated into Vero cells using a single pulse at 450 V and 50 µF with a Bio-Rad Gene Pulser II electroporator. Post-electroporation, the transfected Vero cells were incubated overnight in DMEM supplemented with 1% fetal bovine serum (FBS), followed by further cultivation in FBS-free DMEM. The transfected cells were monitored daily for the development of cytopathic effects (CPE). Recovered viruses underwent plaque purification and were passaged through Vero cells.

### 2.3. Growth Curves of Recombinant Viruses on Vero Cells

Vero cells were infected with IBV-P65 and recombinant IBV and harvested at different times post-infection. Viral stocks were prepared by freezing/thawing the cells three times. The 50% tissue culture infection dose (TCID_50_) of each virus was determined by infecting five wells of Vero cells on 96-well plates in duplicate with 10-fold serial dilution of each viral stock.

### 2.4. Analysis of RNA Synthesis of EP3 by Real-Time PCR

Confluent monolayers of Vero cells in six-well plates were inoculated with IBV-P65 and IBV-EP3 at a multiplicity of infection (MOI) of 1. Following a 1 h period of incubation at 4 °C, the cells were washed twice with phosphate-buffered saline (PBS) and subsequently cultured in 2 mL of serum-free DMEM at 37 °C. Virus-infected cells were harvested at 0, 4, 8, 12, 16, and 24 h post-infection then lysed using TRizol reagent (Invitrogen, USA), and total RNA was extracted according to the manufacturer’s protocol. The RNA samples were resuspended in nuclease-free water, and their concentrations were quantified using a Nanodrop spectrophotometer. Reverse transcription was conducted using Expand Reverse Transcriptase (Roche, Basel, Switzerland) with equal amounts of RNA and specific primers (P-IBV4527F: 5′-_4527_TTTAGCAGAACATTTTGACGCAGA_4551_-3′ and P-IBV4805R: 5′-_4805_TTAGTAGAACCAACAAACACGACAG_4781_-3′). Real-time PCR was performed using the LightCycler FastStart DNA Master SYBR Green I kit, following the manufacturer’s instructions (Roche, Switzerland). Normalized target gene expression levels were recorded for each time point and virus. The GAPDH primers (forward 5′-GTCAAGGCTGAGAACGGGAA-3′ and reverse 5′-AGTGATGGCATGGACTGTGG-3′) were employed to quantify the relative expression levels. The relative expression levels were calculated using the 2^−ΔΔCt^ method, with GAPDH serving as the reference gene. All experiments were performed in triplicate.

### 2.5. Growth Curves of Viruses

To examine the growth kinetics of viruses, confluent monolayers of Vero cells in 96-well plates were inoculated with IBV-P65 and mutant viruses at a MOI of 0.5. Virus-infected cells were collected at 0, 4, 8, 12, 16, 24, and 36 h post-infection. Virus stocks were prepared by subjecting the infected cells to three cycles of freeze–thaw. TCID_50_ was determined for each virus stock. All experiments were conducted in triplicate, and the mean values were calculated.

### 2.6. Western Blotting

Confluent monolayers of Vero cells in six-well plates were inoculated with IBV-P65 and mutant viruses at a MOI of 0.5. Cells were harvested at 0, 4, 8, 12, 16, 24, and 36 h post-infection. At each time point, infected cells were washed with phosphate-buffered saline (PBS), lysed in 2× SDS loading buffer containing 100 mM dithiothreitol (DTT), and heated to 100 °C for 5 min before clarification. Proteins were separated by SDS-PAGE and transferred onto a polyvinylidene difluoride (PVDF) membrane (Stratagene, San Diego, CA, USA). The membrane was blocked overnight at 4 °C or for 2 h at room temperature using blocking buffer (5% fat-free milk powder in PBS containing 0.1% Tween 20). It was then incubated with primary antibodies (anti-S2 at 1:5000; actin at 1:500) diluted in blocking buffer for 2 h at room temperature. Following three washes with PBST, the membrane was incubated with horseradish peroxidase-conjugated secondary antibodies (anti-mouse or anti-rabbit IgG, Dako, Glostrup, Denmark) diluted at 1:2000 in blocking buffer for 1 h at room temperature. After additional washes with PBST, polypeptides were visualized using a chemiluminescence detection kit (ECL kit, Amersham Biosciences, Piscataway, NJ, USA) according to the manufacturer’s protocol. The intensities of the protein bands were quantified using the Image J (1.53e) program.

## 3. Results

### 3.1. Assessment of Infectivity of IBV-EP3 and IBV-P65 in Vero Cells

To compare the infectivity of IBV-EP3 and IBV-P65 in Vero cells, real-time PCR was employed to quantify the synthesis of positive-sense and negative-sense viral RNAs in infected Vero cells. The gene expression levels at each time point were normalized relative to the expression level at 0 h. The normalized gene expression levels of both positive and negative RNA strands are presented in Figure 2. For IBV-EP3, there was no significant increase in RNA levels, with only a 1.12-fold and 1.78-fold increase observed for the positive and negative strands, respectively, at 16 h post-infection. In contrast, the levels of both positive and negative RNA strands in IBV-P65-infected cells progressively increased over time, reaching 169.82-fold and 36.31-fold increases at 16 h (Figure 2A,B). These findings confirm that IBV-EP3 exhibits significantly lower infectivity in Vero cells compared to IBV-P65.

### 3.2. Recovery of Recombinant Viruses

Sequence alignment revealed 19 amino acid variations in the S1 region (1–537 aa) between IBV-P65 and IBV-EP3 [24]. To investigate the functional significance of these mutations in the adaptation of IBV to Vero cells and to identify critical amino acids, six recombinant viruses were generated by substituting the corresponding sequence regions in IBV-P65 with those from IBV-EP3, following the previously established protocol [26]. The characteristics of the resulting recombinant viruses and the associated amino acid changes are summarized in Table 3 (amino acid alignment of the S1 region for both IBV-P65 and IBV-EP3 is shown in Appendix A). Following plaque purification and five consecutive passages through Vero cells, RT-PCR products from the S1 regions were sequenced. Sequence analysis confirmed that no additional mutations occurred beyond the intended amino acid substitutions, indicating the genetic stability of these recombinants in Vero cells.

### 3.3. Growth Characteristics of Recombinant Viruses

Titers of IBV-P65 and six recombinant viruses were determined using the TCID50 method. The growth characteristics of each mutant virus were evaluated by TCID_50_ analysis (Figure 3A,B). As illustrated in these figures, IBV-P65, IBV-21330-21717 (321–451 aa), IBV-20618-20930 (86–180 aa), and IBV-20411-20606 (17–82 aa) exhibited highly similar growth profiles, achieving comparable viral titers at each time point and reaching peak TCID_50_ values at 24 h post-infection. These results suggest that the 14 amino acid substitutions (S38N, Q43H, F56S, S63P, T66I, I69T, H117Y, G118D, Q128K, R156K, K330N, K364S, H391L, D405G) had no significant impact on IBV replication in Vero cells. In contrast, IBV-EP3-S1 (17–451 aa), IBV-20884-21335 (179–323 aa), and IBV-20884-21717 (179–451 aa) displayed markedly delayed and reduced growth phenotypes compared to IBV-P65, with titers approximately 100-fold lower than those of IBV-P65 at 24 h post-infection (Figure 3B). Collectively, these findings indicate that the replacement of the amino acid sequence from positions 179 to 323 in the S1 region of IBV-P65 with the corresponding segment from IBV-EP3 S1 significantly impaired viral growth in Vero cells.

### 3.4. Effects of Amino Acid Changes on Expression of IBV-S

We used Western blot analysis to evaluate the expression of the S proteins of the recombinant viruses. As illustrated in Figure 3, at 16 h post-infection (hpi), the S proteins were clearly detected in Vero cells infected with IBV-P65, IBV-21330-21717, 20,411–20,606, and 20,618–20,930. In contrast, for Vero cells infected with IBV-EP3-S1, 20,884–21,335, and 20,884–21,717, the S proteins were only detectable at 36 hpi, with significantly lower expression levels (Figure 4). These results indicate that the replacement of amino acids 179–323, 179–451, and 17–451 in IBV-P65 with the corresponding amino acids from IBV-EP3 decreased the replication efficiency of IBV in Vero cells.

### 3.5. Effects of Amino Acid Changes on CPE Formation

To visually assess cytopathic effects (CPE) in Vero cells, images were captured at 24 hpi for cells infected with IBV-P65, 20,884–21,335, 20,884–21,717, and 21,330–21,717. The CPEs induced by IBV-P65 and 21,330–21,717 nearly extended across the entire monolayer, whereas those caused by IBV-20884-21717 and 20,884–21,335 were more localized (Figure 5), indicating reduced infectivity.

### 3.6. Amino Acid Sequence Alignment of Different IBV Strains

As shown in Figure 6, the amino acid sequence comparison results demonstrate that threonine (T) was conserved in IBV-M41, EP3, QX-L1148, and 4/91 at the corresponding sites I181 and I273 in S1 of IBV-P65, respectively. Furthermore, T246 in IBV-P65 was also observed in QX-L1148 and 4/91, whereas isoleucine (I) was present at this position in IBV-M41 and EP3. Additionally, cysteine (C) at position 267 in IBV-P65 corresponded to phenylalanine (F) in IBV-M41 and EP3, and leucine (L) in QX-L1148 and 4/91. Moreover, lysine (K) at position 296 in IBV-P65 corresponded to glutamine (Q) in IBV-M41, EP3, and QX-L1148, and histidine (H) in 4/91. Given that IBV-M41 lacks the ability to infect Vero cells and IBV-EP3 exhibits limited replication efficiency in Vero cells, it is likely that the five amino acid changes (T181I, I246T, F267C, T273I, Q296K) play crucial roles in the adaptation of IBV to Vero cells.

In summary, the aforementioned findings indicate that the mutant virus obtained by replacing the 20,884–21,335 nucleotide sequence of the IBV-P65 S1 gene with the corresponding sequence from the IBV-EP3 S1 gene exhibits significantly reduced and delayed growth kinetics, as well as a lower replication efficiency in Vero cells compared to IBV-P65. Considering that only five amino acid substitutions (I181T, T246I, C267F, I273T, K296Q) differentiate this region between IBV-P65 and IBV-EP3, it is plausible to infer that these specific mutations in the S1 domain may enhance the adaptation of IBV to Vero cells.

## 4. Discussion

Field IBV strains generally lack replication capabilities in Vero cells, although some strains are capable of replicating in primary chicken cells, such as chick kidney (CK) cells and chick embryo fibroblasts (DF1). In contrast, the Beaudette strain demonstrates a broader host range, enabling it to replicate efficiently in Vero cells as well as various human and animal cell lines [23,24]. Previous studies have shown that continuous passaging of IBV-EP3 through Vero cells gradually enhances its adaptability to these cells. This process culminates in the generation of IBV-P65 by the 65th passage, a relatively stable strain adapted to Vero and other human and animal cell lines [23,24]. However, the underlying mechanism of this adaptation remains unclear. Sequence comparison revealed that the S1 subunit of the IBV-P65 spike protein contains 19 amino acid mutations relative to IBV-EP3. These mutations are hypothesized to contribute to the improved adaptation. To investigate the effects of these mutations on viral adaptation, in this study, six recombinant mutant viruses were constructed, each harboring partial or complete amino acid substitution in the S1 subunit of IBV-EP3, based on the genetic background of IBV-P65 (Table 3). Our results indicate that five specific amino acid changes (T181I, I246T, F267C, T273I, Q296K) from IBV-EP3 to IBV-P65 S1 enhanced the adaptation of IBV to Vero cells (Figure 3, Figure 4 and Figure 5), while the remaining 14 amino acid changes had no significant effect on IBV’s adaptation to Vero cells. Structurally, 10 conserved amino acids (N38, H43, S56, P63, I66, T69, Y117, D118, K128, and K156) in the S1 units of IBV-EP3 and IBV-M41 are clustered in the S1-NTD region [10]. Although S1-NTD has been reported as the receptor-binding domain (RBD) of the spike protein in IBV M41 and QX-like strains, it contributes to sialic acid binding and plays a critical role in IBV infection, tissue tropism, and pathogenicity [26,27]. Moreover, N38, H43, P63, and T69 in IBV-M41 appear to be critical for binding the spike to the chicken respiratory tract [26]. However, our results indicate that these amino acid changes did not significantly impact the adaptation of IBV-P65 to Vero cells in this study (Figure 3 and Figure 4). Similar results have been reported for recombinant IBV-P65 expressing the S1 region of IBV-H120, as well as the hypervariable regions of QX-like IBV strains [28,29]. These viruses are capable of replicating in Vero cells, suggesting that the acquired ability of IBV to infect Vero cells may not be associated with these specific amino acid changes.

Amino acid sequence alignment demonstrated that the conserved residues T181, I246, F267, T273, and Q296 in IBV-M41 and IBV-EP3 are mutated to I, T, C, I, and K, respectively (Figure 6). IBV-20884-21335, which harbors these specific amino acids, exhibited markedly reduced and delayed growth kinetics as well as diminished replication efficiency in Vero cells compared to IBV-P65 (Figure 3 and Figure 4), suggesting that the specified amino acid substitutions are likely to play pivotal roles in the adaptation of IBV to Vero cells. However, whether a single point mutation or a combination of multiple mutations underlies the adaptability of IBV-P65 to Vero cells remains unclear. Further mapping and detailed analysis of key amino acids will be necessary to elucidate this mechanism. Among the five amino acid residues, I181 in IBV-P65 is located within the NTD, whereas I273 and K296 are situated within the CTD [10]. It remains to be determined whether these amino acid substitutions alter the receptor-binding affinity of the S protein by either directly enhancing receptor interaction or modifying its three-dimensional structure. Further investigation is required to elucidate this mechanism. Determining the three-dimensional structure of this chimeric S protein would provide valuable insights into this issue. Additionally, the IBV S1-CTD functions as an RBD that binds to an unidentified receptor on the surface of chicken cells [10]. Further study is required to determine whether the substitutions T273I and Q296K, located within the CTD, enhance receptor-binding affinity.

N-glycosylation of the IBV-S protein plays a critical role in determining receptor specificity, cellular and tissue tropism, and pathogenicity [30,31,32]. Notably, when employing NetNGlyc-1.0 [33] to predict N-glycosylation sites within the S1 region, it was found that the substitution of threonine at position 273 (T273) in M41 and EP3 to isoleucine at position 273 (I273) in IBV-P65 resulted in the loss of glycosylation at site N271. Further investigation is required to determine whether this loss contributes to enhancing the adaptation of IBV to Vero cells.

Due to their ability to be cultivated in both suspension and adherent cultures, Vero cells facilitate consistent and scalable production of viral yields. As such, they have undergone extensive validation for their efficiency in viral propagation and are officially approved for use in human vaccine production, including the manufacture of polio [34], rabies [35], influenza virus [36], and SARS-CoV-2 vaccines [37]. IBV vaccines, including both live-attenuated and inactivated forms, are currently produced using embryonated hens’ eggs. This process is labor-intensive and costly, primarily because field IBV strains fail to replicate efficiently in cultured cells. Recombinant infectious bronchitis viruses (IBVs) expressing heterologous spike (S) proteins, including either the full S protein or the S1 region derived from the H120, M41, or QX strains, are capable of replicating in Vero cells and confer varying degrees of protection against homologous challenge [7,29,38,39]. These results demonstrate that recombinant rIBVs can effectively express S1 subunits derived from genetically diverse strains of IBV. This functionality will support the rational development of a new generation of IBV vaccines, which have the potential to be propagated in Vero cells. Given that field IBV strains cannot replicate in Vero cells, our findings are expected to facilitate the production of IBV vaccines in Vero cells by enhancing replication efficiency through genetic modification.

## 5. Conclusions

Five amino acid substitutions in the S1 region of the IBV-65 spike protein were identified as critical determinants that enhance the adaptation of infectious bronchitis virus (IBV) to Vero cells. However, the contributions of these substitutions require further investigation.

## Figures and Tables

**Figure 1 vetsci-12-00394-f001:**
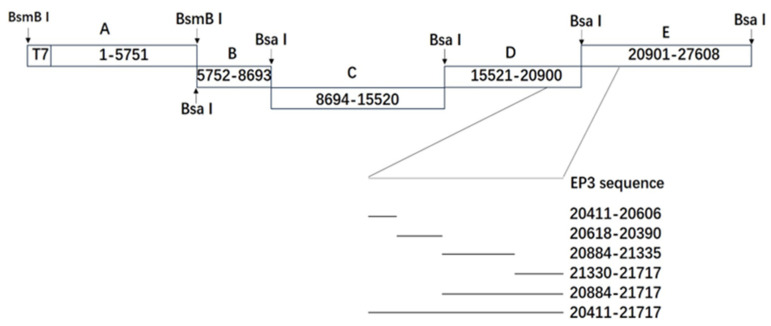
The construction of the recombinant viruses. The figure shows the five RT-PCR fragments of IBV-P65, including the T7 promoter at the 5′-end of fragment A, 30 As at the 3′-end of fragment E, and the replacement regions of EP3-S1 (nucleotides 20,411–21,717).

**Figure 2 vetsci-12-00394-f002:**
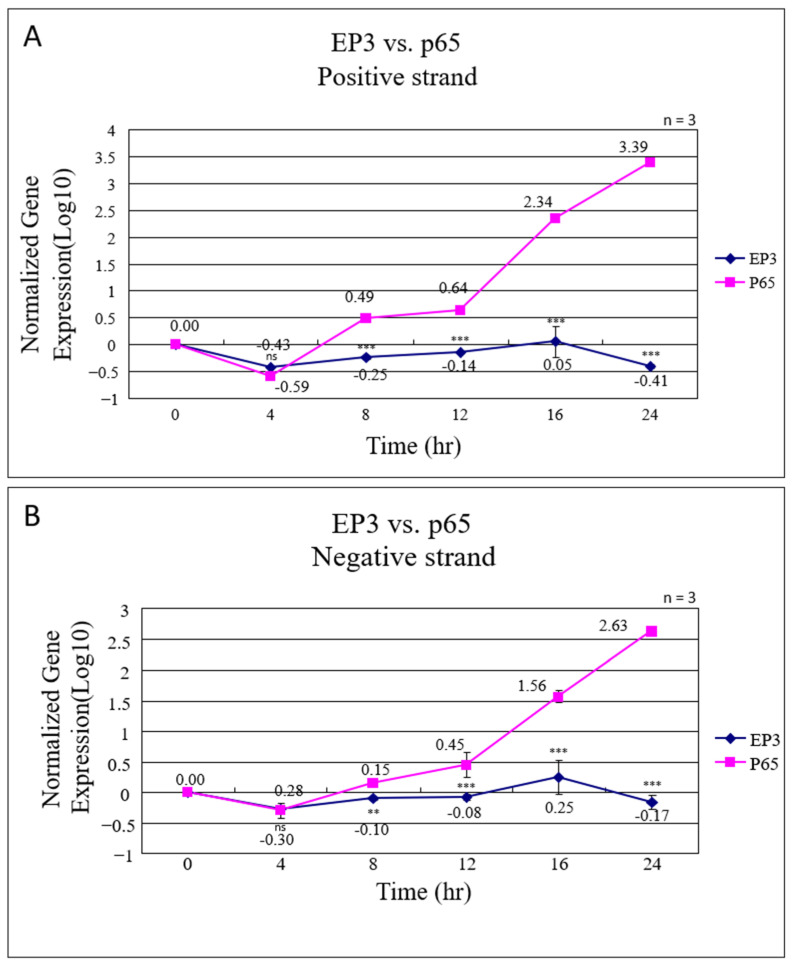
Real-time PCR analysis of positive (**A**) and negative (**B**) viral RNAs from IBV-EP3- and P65-infected Vero cells. Virus-infected cells were harvested at the indicated time points post-infection, and total RNA was extracted for the RT-qPCR analysis of positive and negative RNA synthesis. Normalized target gene expression levels were recorded for each time point and virus. The data represent the mean ± SD from three independent experiments. ns, *p* > 0.05, non-significant; **, *p* < 0.01; ***, *p* < 0.001.

**Figure 3 vetsci-12-00394-f003:**
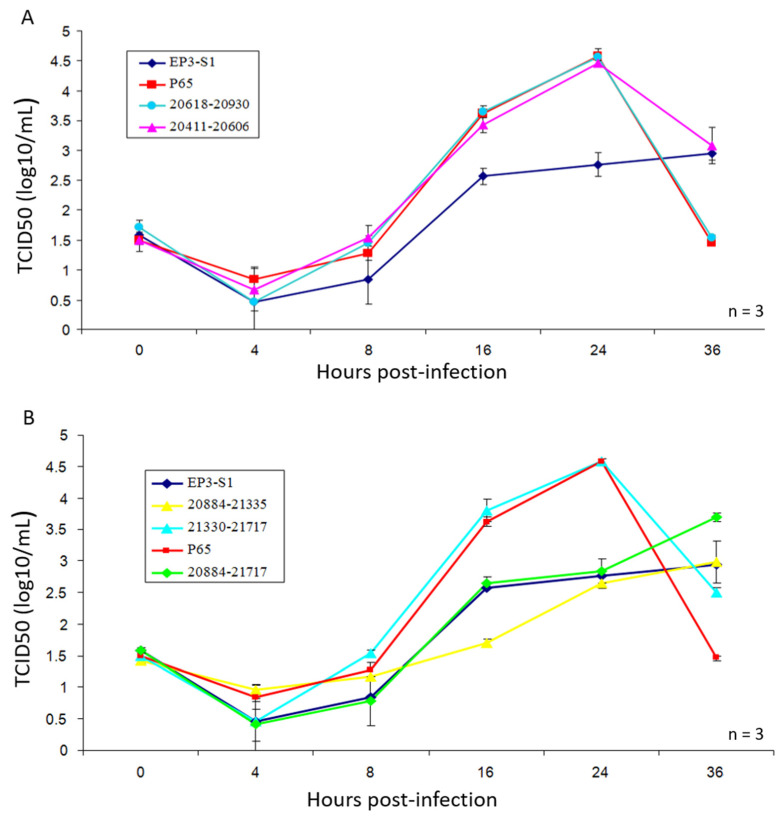
The growth characteristics of IBV-P65 and the recovered recombinant viruses IBV-EP3-S1, 20,618–20,930, and 20,411–20,606 (**A**), as well as 20,884–21,335, 21,330–21,717, and 20,884–21,717 (**B**). The error bars represent the standard error of the mean.

**Figure 4 vetsci-12-00394-f004:**
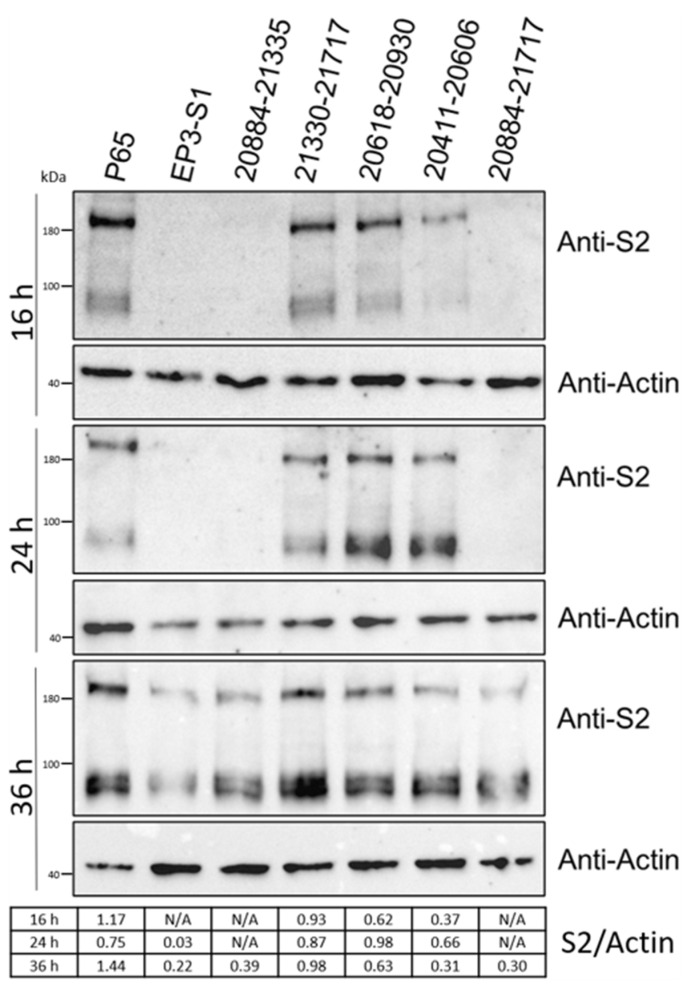
A Western blot analysis of the expression levels of the S protein in Vero cells infected with IBV-P65 and six recombinant viruses. N/A, not applicable.

**Figure 5 vetsci-12-00394-f005:**
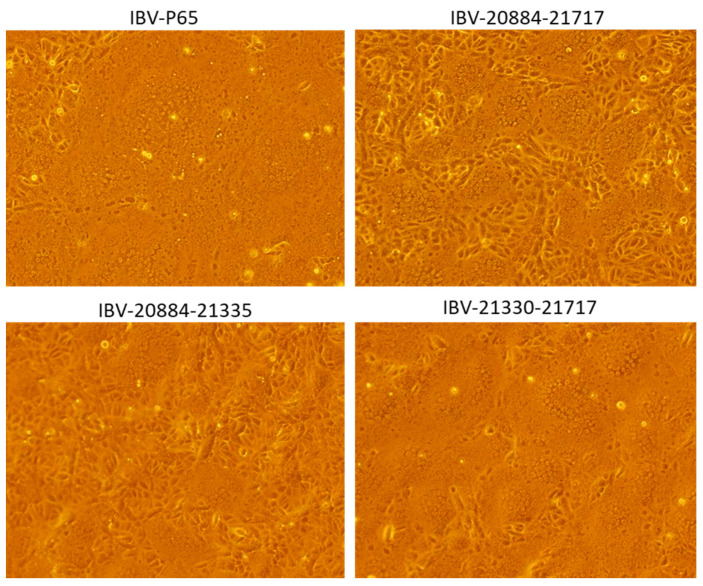
The CPEs induced by IBV-P65 and the recombinant IBV strains (20,884–21,717, 20,884–21,335, and 20,330–21,717).

**Figure 6 vetsci-12-00394-f006:**
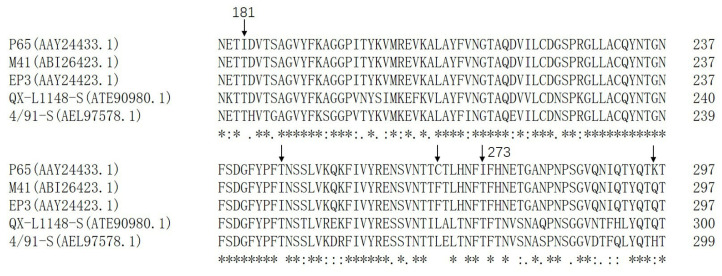
Multiple sequence alignment of the amino acid sequences (179–323 aa) within the S1 region of IBV-P65, IBV-EP3, and IBV-M41. The arrow denotes the position of the mutated amino acid residue. “*”, fully conserved residue; “:”, conserved mutation; “.”, semi-conserved mutation.

**Table 1 vetsci-12-00394-t001:** The primers utilized for the construction of mutant plasmids. The boldface indicates the restriction site of the introduced BsaI enzyme.

Primer	Nucleotide Sequence
BsaI-20411-F	5′-CG**GGTCTC**TATGTAGTGCTGTTTTGTATGA
BsaI-20416-R	5′-CG**GGTCTC**TACATAGTGCACACAAAAGAGT
BsaI-21330-F	5′-C**GGTCTC**GTATGGATCTTATCACCCAA
BsaI-21335-R	5′-C**GGTCTC**CCATACATAAAATTAGACTC
BsaI-20618-F	5′-C**GGGTCTC**CGGCACCGTCATCAGGTAT
BsaI-20623-R	5′-CG**GGTCTC**GTGCCGTCATAGCTATAGA
BsaI-21712-F	5′-CG**GGTCTC**ACCGACTCAGCTGTTAGTTAT
BsaI-21717-R	5′-CG**GGTCT**CGTCGGTTACATTAGTAATAAA
IBV-20884-F	5′-GTTTACACCTCTAATGAGACCATAG
IBV-20930-R	5′-ACACCTGCAGATGTAACATC’
IBV-15511-F	5′-GTGATTCATT**GAGACC**TTTTGC
IBV-15532-R	5′-GCAAAA**GGTCTC**AATGAATCAC
IBV-3′ end	5′-C**ggTCTC**gTTTTTTTTTTTTTTTTTTTTTTTTTTTTTTgCTC

**Table 2 vetsci-12-00394-t002:** The plasmids used in the construction of the mutants.

Plasmid	Region (nt)	Source	Primer	Note
pGEM-IBV-F	15,511–20,416	IBV-P65	IBV-15511-F BsaI-20416-R	Construction of mutant 20,411–20,606
pGEM-IBV-G	20,618–20,930	IBV-P65	BsaI-20618-F IBV-20930-R	
pGEM-EP3-1	20,411–20,623	IBV-EP3	BsaI-20411-F BsaI-20623-R	
pGEM-IBV-H	15,511–20,623	IBV-P65	IBV-15511-F BsaI-20623-R	Construction of mutant 20,618–20,930
pGEM-EP3-2	20,618–20,930	IBV-EP3	BsaI-20618-F IBV-20930-R	
pGEM-IBV-I	21,712–27,608	IBV-P65	BsaI-21712-F IBV-3′ end	Construction of mutant 20,884–21,717
pGEM-EP3-3	20,884–21,717	IBV-EP3	IBV-20884-F BsaI-21717-R	
pGEM-IBV-J	21,330–27,608	IBV-P65	BsaI-21330-F IBV-3′ end	Construction of mutant 20,884–21,335
GEM-IBV-K	20,884–21,335	IBV-P65	IBV-20884-F BsaI-21335-R	
pGEM-EP3-4	21,330–21,717	IBV-EP3	BsaI-21330-F BsaI-21717-R	Construction of mutant 21,330–21,717
pGEM-EP3-5	20,411–21,717	IBV-EP3	BsaI-20411-F BsaI-21717-R	Construction for mutant EP3-S1

**Table 3 vetsci-12-00394-t003:** Detailed information on the recombinant viruses.

Recombinant Viruses	Replaced Nucleotide Sequences (nt)	Replaced Amino Acid Sequences (aa)	Amino Acid Changes from IBV-P65 to IBV-EP3
20,411–20,606	20,411–20,606	17–82	S38N, Q43H, F56S, S63P, T66I, I69T
20,618–20,930	20,618–20,930	86–180	H117Y, G118D, Q128K, R156K
20,884–21,335	20,884–21,335	179–323	I181T, T246I, C267F, I273T, K296Q
21,330–21,717	21,330–21,717	321–451	K330N, K364S, H391L, D405G
20,884–21,717	20,884–21,717	179–451	I181T, T246I, C267F, I273T, K296Q, K330N, K364S, H391L, D405G
EP3-S1	20,411–21,717	17–451	S38N, Q43H, F56S, S63P, T66I, I69T, H117Y, G118D, Q128K, R156K, I181T, T246I, C267F, I273T, K296Q, K330N, K364S, H391L, D405G

## Data Availability

The datasets generated and/or analyzed during the current study are available from the corresponding author on reasonable request.

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
