# Peer review of "Five Amino Acid Substitutions in the S1 Unit of Infectious Bronchitis Virus Are Critical Determinants Enhancing Its Adaptation to Vero Cells"

_vetsci, 2025, doi:10.3390/vetsci12050394_

Round 1
Reviewer 1 Report
Comments and Suggestions for Authors
The manuscript presents a well-structured study on the adaptation of infectious bronchitis virus (IBV) to Vero cells, identifying five amino acid substitutions (I181T, T246I, C267F, I273T, K296Q) in the S1 subunit as critical determinants. However, the justification for selecting these mutations over others remains unclear. Additional discussion on their hypothesized role in adaptation based on structural or sequence homology would strengthen the rationale. Furthermore, a direct comparison with other IBV strains known to infect mammalian cells, such as the Beaudette strain, through sequence alignments, would provide valuable context.
Regarding data presentation, figures should include statistical analyses (e.g., p-values) to validate the significance of differences in viral replication and gene expression.
The methods section requires further details on the reverse genetics system, specifically regarding how recombinant viruses were verified to rule out unintended mutations. Additionally, the growth curve experiments should explicitly state whether multiple biological replicates were performed.
In the discussion, the potential evolutionary implications of these mutations should be explored, particularly in relation to IBV host range expansion beyond Vero cells. Addressing these aspects would significantly enhance the clarity and impact of the study.
Author Response
The manuscript presents a well-structured study on the adaptation of infectious bronchitis virus (IBV) to Vero cells, identifying five amino acid substitutions (I181T, T246I, C267F, I273T, K296Q) in the S1 subunit as critical determinants. However, the justification for selecting these mutations over others remains unclear. Additional discussion on their hypothesized role in adaptation based on structural or sequence homology would strengthen the rationale. Furthermore, a direct comparison with other IBV strains known to infect mammalian cells, such as the Beaudette strain, through sequence alignments, would provide valuable context.
We thank the reviewer for this valuable suggestion. We have incorporated several relevant points raised by the reviewers into the discussion section. For detailed information, please refer to the updated discussion section.
Regarding data presentation, figures should include statistical analyses (e.g., p-values) to validate the significance of differences in viral replication and gene expression.
We thank the reviewer for this important suggestion. In the results section, Figure 2 was updated following the statistical analysis of the data, and detailed explanatory notes were incorporated into the figure legend.
The methods section requires further details on the reverse genetics system, specifically regarding how recombinant viruses were verified to rule out unintended mutations. Additionally, the growth curve experiments should explicitly state whether multiple biological replicates were performed.
We thank the reviewer for this important suggestion. In the Materials and Methods section, we have added a detailed description of the reverse genetics system. A diagram and a table related to recombinant virus construction have also been included. See the Materials and Methods section for details. In addition, three independent biological replicates of the growth curve experiments were conducted, as described in the Materials and Methods section. An n value has also been included in the updated Figure 3 for clarity.
In the discussion, the potential evolutionary implications of these mutations should be explored, particularly in relation to IBV host range expansion beyond Vero cells. Addressing these aspects would significantly enhance the clarity and impact of the study.
We thank the reviewer for this insightful suggestion. In response, we added these elements into discussion and cited relevant references:
“Field IBV strains generally lack replication capabilities in Vero cells, although some strains are capable of replicating in primary chicken cells, such as chick kidney (CK) cells and chick embryo fibroblasts (DF1). In contrast, the Beaudette strain demonstrates a broader host range, enabling it to replicate efficiently in Vero cells as well as various human and animal cell lines [23,24]. Previous studies have shown that continuous passaging of IBV-EP3 in Vero cells gradually enhances its adaptability to these cells. This process culminates in the generation of IBV-P65 by the 65th passage, a relatively stable strain adapted to Vero and other human and animal cell lines [23,24]. However, the underlying mechanism of this adaptation remains unclear. Sequence comparison revealed that the S1 subunit of the IBV-P65 spike protein contains 19 amino acid mutations relative to IBV-EP3. These mutations are hypothesized to contribute to improved adaptation. To investigate the effects of these mutations on viral adaptation, in this study, six recombinant mu-tant viruses were constructed, each harboring partial or complete amino acid substitution in the S1 subunit of IBV-EP3, based on the genetic background of IBV-P65 (Table 3).”
“Due to their ability to be cultivated in both suspension and adherent cultures, Vero cells facilitate consistent and scalable production of viral yields. As such, they have undergone extensive validation for their efficiency in viral propagation and are officially approved for use in human vaccine production, including the manufacture of polio [35], rabies vaccines [36], influenza virus vaccines [37], and SARS-CoV-2 vaccine [38]. IBV Vac-cines, including both live-attenuated and inactivated forms, are currently produced using embryonated hen's eggs. This process is labor-intensive and costly, primarily because field IBV strains fail to replicate efficiently in cultured cells. Recombinant infectious bronchitis viruses (IBVs) expressing heterologous spike (S) proteins, including either the full S protein or the S1 region derived from the H120, M41, or QX strains, are capable of replicating in Vero cells and confer varying degrees of protection against homologous challenge [7,29,39,40]. These results demonstrate that recombinant rIBVs can effectively express S1 subunits derived from genetically diverse strains of IBV. This functionality will support the rational development of a new generation of IBV vaccines, which have the potential to be propagated in Vero cells. Given that field IBV strains cannot replicate in Vero cells, our findings are expected to facilitate the production of IBV vaccines in Vero cells by enhancing replication efficiency through genetic modification.”

Reviewer 2 Report
Comments and Suggestions for Authors
The authors identified the region or amino acid sequences in the IBV S1 protein that are crucial for the adaptation of IBV to Vero cells using the mutated recombinant IBV. The results of this paper are clear and provide answers to the authors' questions. However, this paper still requires some improvements.
Major comments
1. The Beaudette strain and its derivatives were generated through serial passages, and they are not field strains. Therefore, a more detailed background and motivation for this study should be provided regarding the necessity to investigate the adaptation of IBV to Vero cells. It would be also desirable to describe in more detail the significance of this study for future research on IBV.
2. In-depth discussions of the possible effects and mechanisms of the five amino acid sequences on IBV adaptation to Vero cells should be provided.
Minor comments
Title: It is not clear from this study whether all five amino acids are essential for adaptation. Therefore, the title should be improved.
Line 32; spike (S) should be “S”. (Abbreviations are shown above.)
Line 38; infectious bronchitis virus (IBV) should be “IBV”. (Abbreviations are shown above.)
Line 227; “(Fang et al., 2005)” should be shown as a reference number.
Lines 240–241, Figure 5; This should be explained in the Results section.
Author Response
The authors identified the region or amino acid sequences in the IBV S1 protein that are crucial for the adaptation of IBV to Vero cells using the mutated recombinant IBV. The results of this paper are clear and provide answers to the authors' questions. However, this paper still requires some improvements.
Major comments
- The Beaudette strain and its derivatives were generated through serial passages, and they are not field strains. Therefore, a more detailed background and motivation for this study should be provided regarding the necessity to investigate the adaptation of IBV to Vero cells. It would be also desirable to describe in more detail the significance of this study for future research on IBV.
We thank the reviewer for this insightful suggestion. We extend our sincere gratitude to the reviewers for their insightful comments. We concur that this part of the manuscript is of significant importance. In response, we have revised and enhanced the discussion section. We kindly request you to review the updated discussion section.
- In-depth discussions of the possible effects and mechanisms of the five amino acid sequences on IBV adaptation to Vero cells should be provided.
We thank the reviewer for this valuable suggestion. Please find below the details of the changes made, which are also highlighted in red in the Tracked Changes version of the new manuscript.
Revised:
“Amino acid sequence alignment demonstrated that the conserved residues T181, I246, F267, T273, and Q296 in IBV-M41 and IBV-EP3 are mutated to I, T, C, I, and K, respectively (Figure 6). Amino acid sequence alignment revealed that the conserved T181, I246, F267, T273, and Q296 in IBV-M41 and IBV-EP3 is mutated to I, T, C, I, and K, respectively (Figure 6). The IBV-20884-21335, which harbors these specific amino acids, exhibited markedly reduced and delayed growth kinetics as well as diminished replication efficiency in Vero cells compared to IBV-P65 (Figure 3-4), suggesting that the specified amino acid substitutions are likely to play pivotal roles in the adaptation of IBV to Vero cells. However, whether a single point mutation or a combination of multiple mutations underlies the adaptability of IBV-P65 to Vero cells remains unclear. Further mapping and detailed analysis of key amino acids will be necessary to elucidate this mechanism. Among the five amino acid residues, I181 in IBV-P65 is located within the NTD, whereas I273 and K296 are situated within the CTD [10]. It remains to be determined whether these amino acid substitutions alter the receptor-binding affinity of the S protein by either directly enhancing receptor interaction or modifying its three-dimensional structure. Further investigation is required to elucidate this mechanism. Determining the three-dimensional structure of this chimeric S protein would provide valuable insights into this issue. Additionally, the IBV S1-CTD functions as an RBD that binds to an unidentified receptor on the surface of chicken cells [10]. Whether the substitutions T273I and Q296K, located within the CTD, enhance receptor-binding affinity requires further study.
N-glycosylation of the IBV-S protein plays a critical role in determining receptor specificity, cellular and tissue tropism, as well as pathogenicity [31-33]. Notably, when employing NetNGlyc-1.0 [34, 35] for predicting N-glycosylation sites within the S1 region, it was found that the substitution of threonine at position 273 (T273) in M41 and EP3 to isoleucine at position 273 (I273) in IBV-P65 resulted in the loss of glycosylation at site N271. Whether this loss contributes to enhancing the adaptation of IBV to Vero cells requires further investigation”
Minor comments
Title: It is not clear from this study whether all five amino acids are essential for adaptation. Therefore, the title should be improved.
We appreciate the reviewer’s insightful suggestion. However, after thorough deliberation, we regret to inform you that we are unable to accept the comment regarding the revision of the manuscript title, as we believe the current title encompasses the conclusions presented in the manuscript.
Line 32; spike (S) should be “S”. (Abbreviations are shown above.)
We sincerely apologize for the mistake and thank you for your comment. We have revised this detail on Line 49.
Line 38; infectious bronchitis virus (IBV) should be “IBV”. (Abbreviations are shown above.)
We sincerely apologize for the mistake and thank you for your comment. We have revised this detail on Line 55.
Line 227; “(Fang et al., 2005)” should be shown as a reference number.
We sincerely apologize for the mistake and thank you for your comment. We have rewritten this paragraph from Line 288 to Line 306 and cited according to the format required by the journal.
Lines 240–241, Figure 5; This should be explained in the Results section.
Thank you for your comment. We have moved this part to the Results section from Line 264 to Line 279.

Reviewer 3 Report
Comments and Suggestions for Authors
Cai et al. attempt to determine the amino acids in the S1 protein of IBV driving tropism. The authors do not have sufficient data to support their claim of IBV mutant 20884-21335 being the only mutant that may enhance adaptation of IBV to Vero cells. Given that another mutant, IBV 20884-21717 also results in reduced replication in Vero cells and CPE in cells, amino acid changes in this mutant could also be important for enhanced replication in Vero cells. As such, this study needs major additional experimental work, such as introducing permutations of these amino acid changes, to accurately narrow down the amino acids responsible for IBV adaptation in Vero cells.
Comments on the Quality of English LanguageSome sentences are incomplete.
Author Response
Cai et al. attempt to determine the amino acids in the S1 protein of IBV driving tropism. The authors do not have sufficient data to support their claim of IBV mutant 20884-21335 being the only mutant that may enhance adaptation of IBV to Vero cells. Given that another mutant, IBV 20884-21717 also results in reduced replication in Vero cells and CPE in cells, amino acid changes in this mutant could also be important for enhanced replication in Vero cells. As such, this study needs major additional experimental work, such as introducing permutations of these amino acid changes, to accurately narrow down the amino acids responsible for IBV adaptation in Vero cells.
We thank the reviewer for this suggestion. We sincerely apologize for the lack of clarity in describing the sequence positions within the manuscript. To address this, we have incorporated both a figure and a table (Figure 1 and Table 2) to explicitly demonstrate the relationship between the mutation positions of the two viruses under investigation. This additional information clearly indicates that the mutation positions of IBV 20884-21335 are encompassed within those of IBV 20884-21717. We trust that these enhancements will significantly improve the clarity and robustness of the conclusions presented in this manuscript.

Reviewer 4 Report
Comments and Suggestions for Authors
Lines 52-53: A few references can be added.
Lines 62-63: "we generated": Passive voice (sentences) should be used. Passive voice is used as the appropriate language in academic studies. Please take this into consideration.
Lines 65-71: This section must not include the results of this study. The aim of the study must be stated here.
Line 18: "adaptation of IBV‐P65 to Vero cells" Line 71:"adaptation of IBV to Vero cells". Please controlled.
Line 74:DMEM ???
Lines 82, 85, 113, 137:The brands of the kits should be given.
Figure 1: Numerical data do not need to be written. It is clear from the figure.
Line 154: There are 2 figures. Figures must be labeled as "A" and "B".
Line 155: "Confluent monolayers of Vero cells ....time point and virus": These explanations are not appropriate here.
Line 189: ". Confluent monolayers of Vero cells .....error of the mean": These explanations are not appropriate here.
Line 204:"Vero cells were infected at an .....loading control".:These explanations are not appropriate here.
Line 216:"Vero cells were infected ...a microscope.".:These explanations are not appropriate here.
Line 227: (Fang et al., 2005): the journal rules must be used.
Line 231: "we constructed": Passive voice (sentences) should be used.
Figure 5 must be moved to the result section.
Discussion should be enhanced using new references.
Author Response
Lines 52-53: A few references can be added.
We thank the reviewer for this suggestion. We have extended the content of the manuscript, particularly in the discussion section, and cited thirteen additional references.
Lines 62-63: "we generated": Passive voice (sentences) should be used. Passive voice is used as the appropriate language in academic studies. Please take this into consideration.
We sincerely apologize for this and thank you for your comment. We have revised this detail on Line 79.
Line 18: "adaptation of IBV‐P65 to Vero cells" Line 71:"adaptation of IBV to Vero cells". Please controlled.
We sincerely apologize for this and thank you for your comment. We have revised all of them from "adaptation of IBV-P65 to Vero cells" to "adaptation of IBV to Vero cells".
Line 74: DMEM ???
We sincerely apologize for this and thank you for your comment. We have added the “Dulbecco's Modified Eagle's Medium” before “DMEM” on Line 93.
Lines 82, 85, 113, 137:The brands of the kits should be given.
We thank the reviewer for this suggestion. The company and country information of the kits have been added.
Figure 1: Numerical data do not need to be written. It is clear from the figure.
We appreciate the reviewer’s insightful suggestion. We added these numerical data to give readers a more accurate picture of their differences. In addition, we have added statistical analysis to this figure. See Line 193
Line 154: There are 2 figures. Figures must be labeled as "A" and "B".
We sincerely apologize for this mistake and thank you for your comment. We have added "A" and "B" to figure 2. See Line 193.
Line 155: "Confluent monolayers of Vero cells ....time point and virus": These explanations are not appropriate here.
We thank the reviewer for this suggestion. This sentence has been deleted.
Line 189: ". Confluent monolayers of Vero cells .....error of the mean": These explanations are not appropriate here.
We thank the reviewer for this suggestion. This sentence has been deleted.
Line 204:"Vero cells were infected at an .....loading control".:These explanations are not appropriate here.
We thank the reviewer for this suggestion. This sentence has been deleted.
Line 216:"Vero cells were infected ...a microscope.".:These explanations are not appropriate here.
We thank the reviewer for this suggestion. This sentence has been deleted.
Line 227: (Fang et al., 2005): the journal rules must be used.
We sincerely apologize for the mistake and thank you for your comment. We have rewritten this paragraph from Line 288 to Line 306 and cited according to the format required by the journal.
Line 231: "we constructed": Passive voice (sentences) should be used.
We thank the reviewer for this suggestion. We have revised this sentence to passive voice. See Line 307.
Figure 5 must be moved to the result section.
Thank you for your comment. We have moved this part to the Results section from Line 264 to Line 279.
Discussion should be enhanced using new references.
We thank the reviewer for this insightful suggestion. We extend our sincere gratitude to the reviewers for their insightful comments. We concur that this part of the manuscript is of significant importance. In response, we have revised and enhanced the discussion section. Thirteen additional references were cited We kindly request you to review the updated discussion section.

Reviewer 5 Report
Comments and Suggestions for Authors
The manuscript presents a timely and well-conceived study exploring the molecular basis of IBV adaptation to Vero cells using a reverse genetics system. Focusing on five specific amino acid substitutions in the S1 unit represents an important step toward understanding coronavirus tropism and host adaptation. However, to enhance the manuscript’s scientific rigor, reproducibility, and clarity, the following section-specific revisions are recommended:
Abstract and Title
This section appears good. It will be informative if the methods are clearly outlined in the abstract (e.g., qRT-PCR, TCID₅₀, Western blot).
Introduction
This section covers important information regarding the background of the topic. However, it lacks a clearly stated hypothesis or central research question. Furthermore, incorporating more recent and diverse literature could enhance the context.
Materials and Methods
I have just a few minor suggestions for this section.
- Include passage numbers for cell lines and confirm that mycoplasma contamination testing was performed.
- Also, include additional details about mutagenesis, such as the company and country for the primers used, the kit that was utilized, rescue rates, and genome validation.
- No statistical test has been performed. I would recommend performing statistical analyses. Clearly indicate the number of replicates and statistical analyses used (e.g., t-tests, ANOVA, multiple comparisons).
- Incorporate protein loading controls and densitometric quantification for Western blots.
- Detail the microscopy setup and quantify infected cells or fluorescence intensity.
Results
Here are a few suggestions.
- Using statistical comparison would enhance robustness.
- Report how consistently mutant viruses were rescued and confirmed.
- Include normalization of qPCR data using standard curves or reference genes and present units (e.g., genome copies).
- Figure 1 lacks statistical annotations (error bars, significance stars, n-values).
- Is it possible to quantify Western blot bands (e.g., ImageJ) and include a loading control protein?
- For CPE images, apply and elaborate a standardized scoring scale to evaluate severity.
- Include a structural or functional rationale for the impact of identified mutations, referencing spike protein models or similar.
Discussion
Here are a few points that can be addressed.
- Interpretation of how mutations affect receptor binding or tropism is speculative without structural evidence.
- No mention of broader implications (e.g., zoonotic potential, vaccine escape).
- The conclusion does not distinguish between correlation and causation.
Author Response
The manuscript presents a timely and well-conceived study exploring the molecular basis of IBV adaptation to Vero cells using a reverse genetics system. Focusing on five specific amino acid substitutions in the S1 unit represents an important step toward understanding coronavirus tropism and host adaptation. However, to enhance the manuscript’s scientific rigor, reproducibility, and clarity, the following section-specific revisions are recommended:
Abstract and Title
This section appears good. It will be informative if the methods are clearly outlined in the abstract (e.g., qRT-PCR, TCID₅₀, Western blot).
We appreciate the reviewer’s insightful suggestion. Western blot has been added as the detection method of S protein. Indeed, your suggestions will enhance the comprehensiveness of the summary. However, given that the growth characteristics of these viruses include a few of measure methods, listing them individually may lead to redundancy in the sentence structure.
Introduction
This section covers important information regarding the background of the topic. However, it lacks a clearly stated hypothesis or central research question. Furthermore, incorporating more recent and diverse literature could enhance the context.
We thank the reviewer for this valuable suggestion. As mentioned in the section introduction, this manuscript focuses on identifying which part of the 19 changed amino acids influence the growth of IBV in Vero cells. We cited some additional references to broaden the scope of the manuscript, while primarily emphasizing the discussion section.
Materials and Methods
I have just a few minor suggestions for this section.
Include passage numbers for cell lines and confirm that mycoplasma contamination testing was performed.
We thank the reviewer for this valuable suggestion. “Mycoplasma test negatively; passaged three generations after recovery” has been added to Line 92
Also, include additional details about mutagenesis, such as the company and country for the primers used, the kit that was utilized, rescue rates, and genome validation.
We thank the reviewer for this suggestion. The company and country information of the kits and primers have been added.
No statistical test has been performed. I would recommend performing statistical analyses. Clearly indicate the number of replicates and statistical analyses used (e.g., t-tests, ANOVA, multiple comparisons).
We thank the reviewer for this important suggestion. In the results section, Figure 2 was updated following the statistical analysis of the data, and detailed explanatory notes were incorporated into the figure legend.
Incorporate protein loading controls and densitometric quantification for Western blots.
Detail the microscopy setup and quantify infected cells or fluorescence intensity.
We thank the reviewer for this important suggestion. Regarding the protein quantification in Western blot analysis, we used Image J to measure the signal of protein bands after exposure. The results of relative expression levels are presented in the revised Figure 4. The relevant description has been added on Line 176.
Results
Here are a few suggestions.
Using statistical comparison would enhance robustness.
We thank the reviewer for this suggestion. As mentioned above, figure 2 was updated following the statistical analysis of the data, and detailed explanatory notes were incorporated into the figure legend.
Report how consistently mutant viruses were rescued and confirmed.
We thank the reviewer for this suggestion. To ensure successful viral rescue, CPE was monitored on a daily basis following transfection. Subsequently, plaque purification was performed, and the viruses were further passaged in Vero cells.
Include normalization of qPCR data using standard curves or reference genes and present units (e.g., genome copies).
We thank the reviewer for this suggestion. “The GAPDH primers (forward 5’-GTCAAGGCTGAGAACGGGAA-3’ and reverse 5’-AGTGATGGCATGGACTGTGG-3’) were employed to quantify the relative expression levels. The relative expression levels were calculated using the 2−ΔΔCt method, with GAPDH serving as the reference gene” has been added on Line 149.
Figure 1 lacks statistical annotations (error bars, significance stars, n-values).
We thank the reviewer for this important suggestion. As mentioned above, figure 2 was updated following the statistical analysis of the data.
Is it possible to quantify Western blot bands (e.g., ImageJ) and include a loading control protein?
We thank the reviewer for this important suggestion. As mentioned above, we used Image J to measure the signal of protein bands after exposure.
Include a structural or functional rationale for the impact of identified mutations, referencing spike protein models or similar.
We thank the reviewer for this important suggestion. We have included additional information regarding this problem in the section discussion as follows:
“It remains to be determined whether these amino acid substitutions alter the recep-tor-binding affinity of the S protein by either directly enhancing receptor interaction or modifying its three-dimensional structure. Further investigation is required to elucidate this mechanism. Determining the three-dimensional structure of this chimeric S protein would provide valuable insights into this issue. Additionally, the IBV S1-CTD functions as an RBD that binds to an unidentified receptor on the surface of chicken cells [10]. Whether the substitutions T273I and Q296K, located within the CTD, enhance receptor-binding affinity requires further study.
N-glycosylation of the IBV-S protein plays a critical role in determining receptor specificity, cellular and tissue tropism, as well as pathogenicity [31-33]. Notably, when employing NetNGlyc-1.0 [34] for predicting N-glycosylation sites within the S1 region, it was found that the substitution of threonine at position 273 (T273) in M41 and EP3 to isoleucine at position 273 (I273) in IBV-P65 resulted in the loss of glycosylation at site N271. Whether this loss contributes to enhancing the adaptation of IBV to Vero cells requires further investigation.”
Discussion
Here are a few points that can be addressed.
Interpretation of how mutations affect receptor binding or tropism is speculative without structural evidence.
No mention of broader implications (e.g., zoonotic potential, vaccine escape).
The conclusion does not distinguish between correlation and causation.
We thank the reviewer for this important suggestion. With regard to the section discussion, we concur that major revisions are warranted. In the revised manuscript, the updated discussion section has been included.

Round 2
Reviewer 3 Report
Comments and Suggestions for Authors
The authors have significantly improved the manuscript.
Major comments:
Table 3 remains confusing when it comes to comparing the amino acid numbers. It would be best supported with an amino acid alignment of the S1 region of both IBV-P65 and IBV-EP3. Additionally, clarifying which virus’s amino acids are being referred to in columns ‘Replaced amino acid sequence (aa)’ and ‘Amino acids changes from IBV-P65 to IBV-EP3’ would be helpful as well.
Comparison of amino acid 273 for Figure 6 is missing and is important to include.
Author Response
Response to Reviewer comments
Table 3 remains confusing when it comes to comparing the amino acid numbers. It would be best
supported with an amino acid alignment of the S1 region of both IBV‐P65 and IBV‐EP3.
We thank the reviewer for his/her comment. In order to make the comparison of amino acid between
IBV‐P65 and IBV‐EP3 more clearly, we have uploaded Figure S1, which has been indicated on Line 196
and Line 354. We hope this revision will clarify their differences.
Additionally, clarifying which virus’s amino acids are being referred to in columns ‘Replaced amino acid
sequence (aa)’ and ‘Amino acids changes from IBV‐P65 to IBV‐EP3’ would be helpful as well.
We thank the reviewer for this insight comment. The uploaded Figure S1 clearly identifies the amino
acids of the viruses as described above. It is anticipated that this revision will effectively address this
comment.
Comparison of amino acid 273 for Figure 6 is missing and is important to include.
We thank the reviewer for this important suggestion. We sincerely apologize for overlooking this detail,
and the updated Figure 6 now clearly indicates the position of amino acid 273. See Line 259.
